# Trazodone Prolonged-Release Monotherapy in Cannabis Dependent Patients during Lockdown Due to COVID-19 Pandemic: A Case Series

**DOI:** 10.3390/ijerph19127397

**Published:** 2022-06-16

**Authors:** Marianna Mazza, Emanuele Caroppo, Giuseppe Marano, Georgios D. Kotzalidis, Carla Avallone, Giovanni Camardese, Delfina Janiri, Lorenzo Moccia, Alessio Simonetti, Luigi Janiri, Gabriele Sani

**Affiliations:** 1Institute of Psychiatry and Psychology, Department of Geriatrics, Neuroscience and Orthopedics, Fondazione Policlinico Universitario A. Gemelli IRCCS, 00168 Rome, Italy; giuseppemaranogm@gmail.com (G.M.); giorgio.kotzalidis@gmail.com (G.D.K.); avallonecarla@yahoo.it (C.A.); giovanni.camardese@unicatt.it (G.C.); delfina.janiri@gmail.com (D.J.); lorenzomoccia27@gmail.com (L.M.); alessio.simo@gmail.com (A.S.); luigi.janiri@unicatt.it (L.J.); gabriele.sani@unicatt.it (G.S.); 2Department of Psychiatry, Università Cattolica del Sacro Cuore, 00168 Rome, Italy; 3Department of Mental Health, Local Health Authority Roma 2, 00159 Rome, Italy; emanuelecaroppo@gmail.com; 4Department of Neurosciences, Mental Health, and Sensory Organs (NESMOS), Sapienza University of Rome, 00189 Rome, Italy

**Keywords:** trazodone, COVID-19, cannabis use disorder, sleep disorders, anxiety

## Abstract

(1) Background: During the SARS-CoV-2 (COVID-19) pandemic, cannabis use increased relative to pre-pandemic levels, while forced home confinement frequently caused sleep/wake cycle disruptions, psychological distress, and maladaptive coping strategies with the consequent appearance of anxiety symptoms and their potential impact on substance use problems. (2) Aim: Long-acting trazodone (150 mg or 300 mg daily) has a potential benefit as monotherapy in patients with cannabis use disorder. The present work aims to investigate the effectiveness of trazodone in optimizing the condition of people with cannabis dependence under pandemic conditions. (3) Methods: All cases with cannabis use disorder were uniformly treated with long-acting trazodone 150 mg or 300 mg/day; their craving and clinical status were monitored through appropriate psychometric scales. Side effects were recorded as they were reported by patients. We described the cases of three young patients—one man and two women—who were affected by chronic cannabis use disorder and who experienced lockdown-related psychological distress and sought psychiatric help. (4) Results: The described cases highlight that the once-a-day formulation of trazodone seems to have a therapeutic role in patients with cannabis use disorder and to guarantee tolerability and efficacy over time. No significant side effects emerged. (5) Conclusions: The use of long-acting trazodone (150 mg or 300 mg daily) has a potential benefit as monotherapy in patients with cannabis use disorder. Trazodone deserves to be studied in terms of its efficacy for cannabis use disorder.

## 1. Introduction

During the SARS-CoV-2 (COVID-19) pandemic, profound physical and mental health effects on populations around the world have been described [1]. Some studies outlined that alcohol use and cannabis use increased relative to pre-pandemic levels, with women reporting a greater increase in cannabis use than men [2,3]. Despite its efficacy in limiting the number of infections, forced home confinement frequently caused sleep/wake cycle disruptions, psychological distress, and maladaptive coping strategies [4] associated with the onset of anxiety symptoms and their potential impact on substance use problems during the COVID-19 crisis [5]. It has been acknowledged that the neurophysiology of addiction and sleep disorders share common neurophysiological mechanisms, resulting in a mutual interaction [6]. Diagnosing and treating primary sleep disorders, particularly in young patients, can prevent the development of addiction in susceptible individuals. In particular, insomnia is a typical characteristic of cannabis withdrawal syndrome and a primary cause of relapse in cannabis use disorder. An ideal sleep aid should prevent relapse and have low abuse potential [6].

Trazodone is a triazolopyridine derivative that works by inhibiting both serotonin transporter and serotonin type 2 receptors and is included in the category of SARI drugs (serotonin antagonist and reuptake inhibitors) [7]. It suppresses raphe serotonergic neuronal firing through partial 5-HT_1A_ agonism and alpha1-adrenoceptor inhibition [8]. Administered chronically, like other antidepressants, it downregulates beta-adrenocpetors in the brain [9]. All these actions are believed to underlie its antidepressant actions. Furthermore, it blocks histamine H-1 receptors at low doses [10]. This action is believed to underpin its sleep-inducing action. It also has some affinity for sigma_2_ receptors, although the nature and significance of this affinity is not entirely clear [11]. It is an FDA-approved drug for the treatment of major depression, in combination or alone. Trazodone is also used off-label for sleep disorders, anxiety, behavioral disorders associated with dementia and Alzheimer’s disease, bulimia, sexual dysfunction, fibromyalgia, obsessive-compulsive disorder, and post-traumatic stress disorder (PTSD) [12]. Although trazodone is approved and marketed in most countries worldwide for the sole treatment of major depressive disorder, not only is the use for this medication is very common for many other conditions, but other—not officially approved—uses of trazodone include the treatment of chronic pain, diabetic neuropathy, and benzodiazepine and/or alcohol dependence or abuse. In addition, due to its 5HT_2A_ receptor antagonistic action, trazodone may be used to prevent the occurrence of initial and long-term side effects of selective-serotonin-reuptake inhibitors (SSRI), such as anxiety, insomnia, and sexual dysfunction [13].

Prolonged release once-a-day formulation of trazodone (150 mg and 300 mg bisectable tablets) is able to control the release of trazodone over 24 h and was developed in an attempt to enhance patient compliance to therapy without a loss in efficacy, as well as to improve tolerability by avoiding the early high peak plasma concentration seen with conventional formulations [14].

## 2. Case Reports

### 2.1. Method

Based on the literature described above, we decided to treat patients referred to our service with prolonged release once-daily trazodone in 150 mg oral bisectable tablets, and to increase it to 300 mg according to needs. For treatment, we used the algorithm in Figure 1. We obtained approval for treatment from our Ethical Committee and from patients and provided full information on the treatment’s nature and effects, including possible adverse events, which we encouraged patients to report, although we recommended that the patient should not focus upon them. We provided the patient with a visual analogue scale (VAS) to assess craving which consisted of a 10 cm bar with an extreme end marked as “none” and the other “as high as I have ever experienced”. We asked patients to make a mark on the form each day. At weekly return visits, we assessed patients with appropriate psychometric scales (Hamilton Depression Rating Scale and Hamilton Anxiety Rating Scale). Treatment was agreed upon with the patient at each visit, and if an unsatisfactory response was obtained, we could propose increasing the dosage to 300 mg/day. At control visits, we could offer psychotherapy sessions to patients who could accept them. Ethical standards of the World Medical Association (Helsinki Assembly on Human Rights) were adhered to. Data were anonymized through feeding data to an anonymous datasheet and privacy requirements respected through rendering subjects unrecognizable through their stories. All patients filled out the Consent Form after being assessed for capacity for providing treatment consent. They all actively participated in the treatment decision process and agreed all received treatments with their caring physicians. They signed the standard Consent Form after being fully informed on the nature of their condition, on the need to receive treatment for their condition, and on the nature of the treatments they were going to receive. They all provided consent for publication of their cases. The treatment protocol was approved by the Ethical Committee of the Fondazione Policlinico Universitario Agostino Gemelli IRCCS—Università Cattolica del Sacro Cuore, Rome, with the protocol number Prot. ID 3275. All treatments respected the Principles of Human Rights, as adopted by the World Medical Association at the 18th WMA General Assembly, Helsinki, Finland, June 1964 and subsequently amended at the 64th WMA General Assembly, Fortaleza, Brazil, October 2013. Patients were informed and accepted that their case would receive publication in an academic journal and that they had the right to refuse publication and to withdraw from treatment at any time.

### 2.2. Case 1

A 31-year-old man with 10 years of cannabis use disorder and occasional use of cocaine called the psychiatric emergency telephone line during the lockdown period, complaining a worsening of sleep disorder, particularly middle insomnia that was treated in the past with various benzodiazepines, melatonin, and zolpidem prescribed by a private psychiatrist. However, all these remedies for sleep disorder failed to yield any substantial benefit. The psychiatrist had also prescribed 20 mg of paroxetine per day which the patient had to stop after one month for QTc prolongation on the electrocardiogram (ECG). The patient developed no paroxetine discontinuation related symptoms upon abrupt suspension.

The patient was interviewed in an outpatient setting. He had no family history of anxious or mood disorders nor did he complain of symptoms of an anxious or depressive type. He reported having a job as a clerk in a financial company. He described his family, peer, and community relationships as strong. He reported smoking marijuana once or twice daily. He had always suffered from insomnia from a very young age. Since his adolescence, he reported that the use of cannabis helped him sleep better. Recently, during the lockdown, his sleep disturbances worsened and it seemed that no prescription could be successful. During the day he felt calm but very fatigued and this was beginning to create problems for him at work. Due to this, he decided to switch to working from home. Besides this, he began to worry that prolonged use of cannabis had impaired his cognitive abilities, although he could not document this through objective facts.

The patient was prescribed 150 mg prolonged-release trazodone once daily in the evening. After two weeks of continuous improvement, the patient showed a restored sleep pattern without adverse effects. In particular, he began to fall asleep more quickly and to no longer show nocturnal awakenings. At the same time, he managed to completely suspend cannabis use without manifesting craving or withdrawal phenomena. He agreed to begin a psychodynamic therapy, which he practiced within a group setting. At week 32 after being seen, the patient was continuing his prescribed treatment and was still abstaining from the use of cannabis. No ECG abnormalities have been observed such as those the patient developed when he was taking paroxetine.

### 2.3. Case 2

A 19-year-old woman was admitted to our Emergency Unit with cannabinoid hyperemesis syndrome (CHS), which is characterized by cyclic vomiting and nausea caused by chronic cannabis usage [15]. She disclosed 5 years of cannabis use disorder confirmed by urine toxicology screening. The patient reported trying to resolve the addiction two years earlier by undergoing a course of cognitive-behavioral psychotherapy for six months with no apparent benefit. During the lockdown due to COVID-19, she was forced to attend school through the distance learning method. During this time, she reported missing her friends very much; she also reported a worsening of her academic performance. She admitted an increase in substance use associated with psychosocial distress and described this behavior as an attempt to self-medicate her unease and manage her mental health amidst the pandemic.

CHS is often refractory to the standard treatment for nausea and vomiting [16]. Unconventional antiemetics, such as haloperidol, have proved to be successful in alleviating symptoms; however, they can lead to adverse effects, such as dystonia, extrapyramidal reactions, and in rare cases, even to severe clinical pictures, such as neuroleptic malignant syndrome [17]. The patient reported she was previously prescribed 1 mg oral haloperidol in a hospital setting, but had to discontinue shortly after due to the development of adverse effects. We treated her for 48 h with low doses of trazodone (100 mg/day) by i.v. infusion. This was followed by a benign clinical course and resolution of symptoms. At discharge, she consented to taking 150 mg of prolonged-release trazodone once daily in the evening for two weeks, and to increase it subsequently to 300 mg a day. The patient said she no longer had cravings for cannabis. She also immediately felt more relaxed and more focused on her academic activities. She was also advised to initiate a psychotherapy, but she refused. However, she agreed to undertake an online psychoeducation counseling course. At the 28^th^ week after her first visit, the patient was still continuing the prescribed therapy and reports she was abstaining from cannabis use. The patient reported no side effects while on trazodone. Laboratory examinations revealed no alterations or off-normal values.

### 2.4. Case 3

A healthy 24-year-old woman with a 6-year history of regular cannabis use presented with chest pain on the left side and dizziness for 6 h. She was admitted to a psychiatric day hospital and the investigation of cardiological measures showed normal troponin T levels and a normal ECG. She was living alone during the lockdown and reported anxiety, fatigue, and insomnia in the week prior to admission. She was referred by the general practitioner for psychological counseling. Her family and personal histories were free from psychiatric disorders; she reported that she had never taken psychiatric drugs. She also reported that she had always used cannabis on a daily basis alone or in company to feel more comfortable in social relationships and at work, as being a saleswoman in a department store she was subjected to intense shifts. With the lockdown and the closure of the shops, her daily life received a sudden severe limitation, which she described as “emptiness”. This was enhanced by the fact that she no longer had the opportunity to reach her family, because her close family members were living in another region of Italy. It should be reminded that during the lockdown, It was forbidden to cross regional borders. She had recently ended a romantic relationship and this had increased her sense of loneliness. During the forced stay at home, she had noticed an increase in her food intake and a more frequent alcohol intake. She reported she was eating normally before the pandemic and that she assumed alcohol only sporadically. Daily cannabis use remained unchanged. However, she had noticed that the usual quantity of smoked cannabis failed to produce the relaxing effect she previously achieved.

The patient agreed with her physicians to take 150 mg prolonged-release trazodone once daily in the evening; this was followed by a rapid resolution of her symptoms. She continued to attend psychological counseling sessions. At the 27th week after she was first seen at our service, the patient was continuing her therapy and this was followed by symptom resolution and associated with abstinence from cannabis use as well as alcohol intake. No ECG abnormalities were observed at the 27-week follow-up.

## 3. Discussion

We described the cases of three young patients—one man and two women—who were affected by chronic cannabis use disorder and who experienced lockdown-related psychological distress and sought psychiatric help. We treated them all with a 150 mg prolonged-release trazodone dose once daily in the evening and this was followed by prompt symptom resolution accompanied by abstinence from cannabis use; benefits were maintained in all cases for the entire follow-up period, which ranged from 27 to 32 weeks according to case. Trazodone is an antidepressant that inhibits the reuptake of serotonin and blocks the histamine and alpha-1-adrenergic receptors. It also induces significant changes in presynaptic receptor adrenoreceptors on 5-HT central neurons. The full spectrum of trazodone’s mechanism of action is not fully understood, which could explain its off-label uses [12]. Thanks to its antihistaminic H-1 activity, trazodone is increasingly used in patients with sleep disorders, especially in the elderly, also for its favorable tolerability profile [18].

Although trazodone is used in alcohol use disorder and reduces alcohol craving [19], and in rapid opioid detoxification (methadone withdrawal) in conjunction with naltrexone has shown similar efficacy to that of clonidine [20], it has not been trialed in cannabis use disorder. The case of the efficacy of trazodone in alcohol withdrawal syndrome is difficult to explain, but its neurobiology mainly involves the NMDA glutamatergic receptors and the GABAergic GABA_A_ receptors [21,22,23], and not the cannabinoid receptors. The cannabinoid receptors were discovered during the 1980s and their endogenous ligand, anandamide, was isolated in 1992 [24]. There are two types of cannabinoid receptors, CB_1_ and CB_2_, the first of which is widely distributed in the brain [25]. Cannabis mainly acts through CB_1_ receptors [26]. While there is basic science support to the use of trazodone in opioid withdrawal syndrome [27], and a possible involvement of opioid receptors has been hypothesized in its hypnotic action [28], no interference with cannabinoid mechanisms has been described heretofore for trazodone; hence, we are unable to explain the anti-cannabinoid craving effect it showed in all three cases we here reported. However, a variety of drugs used to treat psychiatric disorders comorbid with substance use disorders have shown anti-craving effects [29,30,31], the mechanism of which is unknown. To our knowledge, this is the first report of the use of trazodone in people with cannabis use disorder. Some recent studies suggested that selective serotonin reuptake inhibitors, serotonin–norepinephrine reuptake inhibitors, and the SARI classes of antidepressants drove this protective effect on COVID-19 infection risk [32,33]. Furthermore, it has been shown that during the COVID-19 pandemic—specifically in the 13 weeks following March 4, 2020—there were increased prescriptions in adult patients for trazodone, and decreased prescriptions for benzodiazepines and hypnotics [34]. This may reflect a shift in clinicians’ attitudes and increased awareness of trazodone’s actions.

Trazodone was developed in Italy during the early 1970s [35]. Its first indication was depression [36]. It received FDA approval in 1981 and by 2019 gained the 25th position among the most prescribed antidepressants, maintaining between 20 million and 30 million prescriptions per year for the seven-year period from 2013 to 2019 [37]. Its principal indication is major depression, but the drug proved to be effective in all depressive subtypes [38]. Trazodone is associated with few and rare side effects, with the most notable being priapism [39]. Its favorable adverse effect profile rendered it suitable for use in the elderly and among those with heart problems and insomnia, as its sleep-inducing properties were discovered. The use of trazodone is greater in people with depressive disorders; older age, self-reported sleep problems, and having a nonsubstance use and nonpersonality disorder psychiatric diagnosis are associated with trazodone use [40]. The drug is profitably used in alcohol and other psychoactive substance detoxification clinics [41]. In people recovering from alcohol use disorder and the related sleep problems, trazodone proved not to affect long-term alcohol outcomes [40]. This is in line with the outcome of our Case 3, who abstained from both cannabis and alcohol. Trazodone is widely prescribed as a sleep aid in persons with addictions because of its lack of addictive potential. Cannabis may improve subjective sleep complaints, particularly when used over short periods of time, while chronic cannabis use is associated with negative subjective effects on sleep (poor sleep quality, insomnia, nightmares) that are manifested most prominently during withdrawal [42]. Insomnia disorder impairs quality of life and is associated with an increased risk of physical and mental health problems including anxiety, depression, drug and alcohol abuse, and increased health service use. Hypnotic medications (e.g., benzodiazepines and ‘Z’ drugs) are licensed for sleep promotion, but can induce tolerance and dependence, although many people remain on long-term treatment. Trazodone has demonstrated a moderate improvement in subjective sleep outcomes over placebo with few adverse effects with trazodone than placebo (i.e., morning grogginess, increased dry mouth, and thirst) [43].

Fatal arrhythmias can occur with trazodone overdose [44]. It has been noticed that trazodone—along with cyclobenzaprine and quetiapine—can be used for the treatment and/or self-treatment of opioid withdrawal symptoms, so clinicians need to prescribe these medications to their patients with appropriate caution [45].

In addiction disorders, trazodone is usually used in association with other drugs. Here we showed that, when used at the appropriate dosage and using the appropriate formulation, trazodone may be useful in treating young people with cannabis use disorder as monotherapy. Trazodone demonstrated an ability to counteract substance use and to maintain wellbeing.

## 4. Conclusions

The described cases highlight that long-acting trazodone (150 mg or 300 mg daily) has a potential benefit as monotherapy in patients with cannabis use disorder. The once-a-day formulation of trazodone seems to guarantee improved tolerability and efficacy over time. Trazodone deserves more studies in terms of its efficacy for cannabis dependence. Large, randomized, and controlled clinical trials should be conducted in the near future to evaluate if such an indication could be supported by strong scientific evidence.

## Figures and Tables

**Figure 1 ijerph-19-07397-f001:**
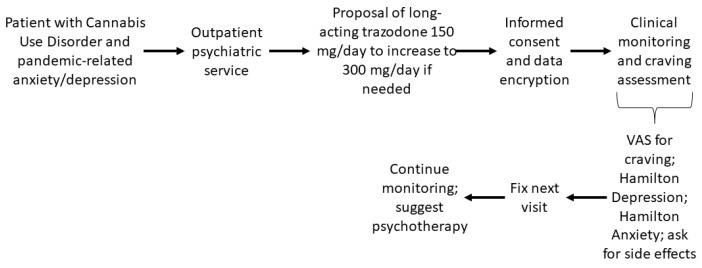
Clinical algorithm followed in evaluating each patient.

## Data Availability

The data presented in this study are available on request from the corresponding author.

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
