# Peer review of "Trazodone Prolonged-Release Monotherapy in Cannabis Dependent Patients during Lockdown Due to COVID-19 Pandemic: A Case Series"

_ijerph, 2022, doi:10.3390/ijerph19127397_

Round 1

Reviewer 1 Report

Abstract should be rearranged in the whey to have background, aim, methods, result and conclusion. Even this is case report.

I think you should add aim of the article and method used to collect data

Author Response

Thank you for this suggestion. We adhered to your indications and modified the Abstract accordingly.

Reviewer 2 Report

I congratulate the authors for the work done. I am grateful with the editors for the possibility of revising this manuscript. Although the quality of the manuscript is high, I would like to make a contribution, please explain the methodology of study in the abstract.

The use of longacting trazodone (150 mg or 300 mg daily) has a potential benefit as monotherapy in patients with cannabis use disorder.

Author Response

Thank you for your kind comments. The Abstract has been modified according to your indications.

Reviewer 3 Report

This article focuses on an important and relevant topic and describes the unique case studies of Trazodone for helping cannabis dependent people with emotional disorders. The article well substantiates the background and explanation of the use of Trazodone from a neurophysiological point of view. 

The study aims to theoretically substantiate and experimentally validate the effectiveness of Trazodone in optimizing the condition of people with cannabis dependence under pandemic conditions.

This topic is important because, first, it is about finding ways to reduce addiction to cannabis. Secondly, the topic is related to finding ways to help people with various psychological and psychophysiological disorders.

This article makes a significant contribution to the area being developed. Such studies of Trazodone have not been conducted before, and the authors show real cases where such therapy has been confirmed.

In general, the methodology of the study is clear. I recommend that the authors add a description of the algorithm for selecting a therapy regimen.

As a wish, I can recommend the authors to describe in more detail how exactly to select the dose and period of treatment. Can this be predicted in advance based on some input data?

I also found two typos in the text:

1. line 41: "is" should be deleted;

2. line 130: "EGC" instead of "ECG".

Author Response

This article focuses on an important and relevant topic and describes the unique case studies of Trazodone for helping cannabis dependent people with emotional disorders. The article well substantiates the background and explanation of the use of Trazodone from a neurophysiological point of view. The study aims to theoretically substantiate and experimentally validate the effectiveness of Trazodone in optimizing the condition of people with cannabis dependence under pandemic conditions. This topic is important because, first, it is about finding ways to reduce addiction to cannabis. Secondly, the topic is related to finding ways to help people with various psychological and psychophysiological disorders. This article makes a significant contribution to the area being developed. Such studies of Trazodone have not been conducted before, and the authors show real cases where such therapy has been confirmed. In general, the methodology of the study is clear.

Response: We thank you for having appreciated our manuscript.

I recommend that the authors add a description of the algorithm for selecting a therapy regimen. As a wish, I can recommend the authors to describe in more detail how exactly to select the dose and period of treatment. Can this be predicted in advance based on some input data?

Response: We thank you for your observation. We produced an algorithm of the procedures used. Dose started at 150 mg/day and titrated until reaching a satisfactory result. It was maintained as long as the patient was finding benefits and wished to remain on the drug. We could not identify predictors, since we treated only three cases up to now.

I also found two typos in the text:

  1. line 41: "is" should be deleted;
  2. line 130: "EGC" instead of "ECG".

Response: Typos have been corrected. Thank you again for your positive and constructive attitude.